# POINTACL: POINT CLOUD UNDERSTANDING VIA ATTENTION-DRIVEN CONTRASTIVE LEARNING

## ABSTRACT

Recently Transformer-based models have advanced point cloud understanding by leveraging self-attention mechanisms, however, these methods often overlook latent information in less prominent regions, leading to increased sensitivity to perturbations and limited global comprehension. To solve this issue, we introduce **PointACL**, an attention-driven contrastive learning framework designed to address these limitations. Our method employs an attention-driven dynamic masking strategy that guides the model to focus on under-attended regions, enhancing the understanding of global structures within the point cloud. Then we combine the original pre-training loss with a contrastive learning loss, improving feature discrimination and generalization. Extensive experiments validate the effectiveness of PointACL, as it achieves state-of-the-art performance across a variety of 3D understanding tasks, including object classification, part segmentation, and few-shot learning. Specifically, when integrated with different Transformer backbones like Point-MAE and PointGPT, PointACL demonstrates improved performance on datasets such as ScanObjectNN, ModelNet40, and ShapeNetPart. This highlights its superior capability in capturing both global and local features, as well as its enhanced robustness against perturbations and incomplete data.

## 1 INTRODUCTION

Point clouds are widely applicable in fields such as robotics (Chen et al., 2020; Tan et al., 2001), autonomous driving (Chen et al., 2017; 2020), augmented reality (Arena et al., 2022), and virtual reality (Garrido et al., 2021) as a representation of objects in three-dimensional space. These diverse applications highlight the significance of obtaining detailed and insightful 3D representations. Despite their potential, the irregular and sparse nature of point cloud data poses significant challenges to precise and efficient 3D processing and understanding.

Recent advancements in deep neural networks, especially Transformer-based models (Pang et al., 2022; Chen et al., 2024; Yu et al., 2022) employing self-supervised learning, have shown promise in point cloud understanding. These models leverage the attention mechanism to capture complex relationships between point patches, prioritizing critical regions for understanding the point cloud while downplaying less significant areas. Originally designed for natural language, attention mechanism has been successfully adapted for 2D vision. However, unlike natural language (Devlin, 2018) or images (He et al., 2022), which often contain redundant information such as contextual structures and backgrounds, point cloud data are inherently sparse, meaning that each point or region is critical to the overall representation. This scarcity of redundant information implies that Transformer-based models, when neglecting less prominent point patches, may inadvertently overlook essential latent information. This observation leads us to a pivotal question: *Can we design a framework that leverages latent information from the global regions of point clouds?*

To answer this question, we re-examine the attention weights in Transformer-based point cloud models. As illustrated in Figure 1, we find that models like Point-MAE (Yu et al., 2022) and Point-GPT (Chen et al., 2024) primarily rely on a limited set of high-attention patches for analysis. This reliance presents two significant issues: **(1)** Increased sensitivity to perturbations. Over-focusing on high-attention patches makes the models more susceptible to noise and incomplete data, as disturbances in these areas disproportionately affect performance. **(2)** Limited global understanding.

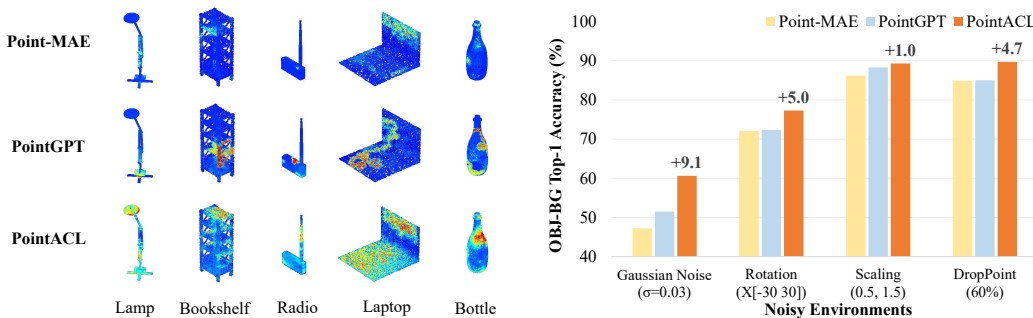

Figure 1: **Illustration of PointACL's Advantages.** Point-MAE is employed as the backbone of our proposed PointACL. **Left:** PointACL emphasizes extracting global information from a greater number of patches. **Right:** PointACL demonstrates greater robustness than previous methods.

Ignoring potential information in low-attention patches constrains the model's ability to develop a comprehensive understanding of the point cloud's global structure.

To solve these issues, we introduce **PointACL**, an **A**ttention-driven **C**ontrastive **L**earning framework for point clouds that can be seamlessly integrated into existing Transformer-based models. Our approach comprises two key components: First, an attention-driven dynamic masking strategy is proposed that aims to mitigate the model's reliance on a limited subset of key patches by guiding it to focus on under-attended regions. Specifically, we construct a dynamic masking probability based on the latest self-attention significance scores, prioritizing masking the patches that contribute most to the global feature representation. This strategy encourages the model to infer global features from less prominent patches, thus fostering a more comprehensive and robust understanding of the point cloud. Furthermore, we combine the original pre-training loss with a contrastive learning objective. It allows the model to retain its task-specific learning capabilities while enhancing its global understanding and generalization through contrastive learning. Compared to previous methods, our approach better captures the global structure of point clouds rather than focusing solely on local features. Consequently, under various noisy environments such as Gaussian noise, rotation, scaling, and point dropout, PointACL significantly enhances the model's robustness.

Our PointACL achieves state-of-the-art performance across various 3D understanding tasks. Specifically, for object classification, PointACL attains accuracies of 89.9% on the challenging PB-T50-RS setting of ScanObjectNN and 94.1% on ModelNet40, with its performance advantage persisting even when competing models are allocated additional training time. In few-shot learning, it sets new benchmarks across all evaluation tasks. Moreover, PointACL demonstrates enhanced robustness against perturbations and incomplete data, consistently outperforming previous approaches under various noisy environments such as Gaussian noise, rotation, scaling, and point dropout. These results highlight PointACL's potential to effectively address the limitations of existing Transformer-based models by capturing comprehensive global structures and fine-grained local details.

Our main contributions can be summarized as follows: **(I)** We propose PointACL, a novel framework that combines self-attention mechanisms with contrastive learning for point cloud understanding which enhances the model's ability to capture global structures and significantly improves its robustness and generalization capabilities. **(II)** We propose an attention-driven dynamic masking strategy that encourages the model to focus on under-attended regions, ensuring learning from diverse patches rather than over-relying on a small subset. **(III)** Extensive experimental results demonstrate that PointACL can be seamlessly integrated into mainstream transformer architectures and achieve significant improvements across a variety of 3D understanding tasks.

## 2 RELATED WORKS

**Self-Supervised Learning for NLP and Image.** Self-supervised learning (SSL) has emerged as a powerful paradigm in natural language processing (NLP) (Erhan et al., 2010; Zhu et al., 2023b) and computer vision (Radford, 2018; Goodfellow et al., 2020; Yu et al., 2017; Misra & Maaten, 2020; Qian et al., 2021; Abdelfattah et al., 2024; Liang et al., 2024), enabling models to learn rich representations from unlabeled data. The core idea is to design pretext tasks that encourage

models to capture underlying data structures. In NLP, BERT (Devlin, 2018) exemplifies this by randomly masking input tokens and training the model to predict them, fostering deep contextual understanding. ELMo (Sarzynska-Wawer et al., 2021) utilizes bidirectional LSTMs to generate contextualized word embeddings, while GPT (Radford, 2018) adopts an autoregressive approach with a unidirectional Transformer to predict the next word, fine-tuning all parameters for specific tasks. In computer vision, contrastive learning initially dominated SSL for images, focusing on grouping similar (augmented) images closer and pushing dissimilar ones apart in the feature space. However, recent generative SSL methods have begun to outperform contrastive approaches. Masked Autoencoders (He et al., 2022) randomly mask a significant portion of image patches and train the model to reconstruct the missing pixels, leading to effective visual representations. BEiT (Bao et al., 2021) extends this by tokenizing image patches and predicting masked tokens, integrating NLP techniques into vision tasks. Additionally, Image GPT (Luppino et al., 2021) treats images as sequences of pixels and trains a Transformer to autoregressively predict pixels without explicit spatial structure, demonstrating strong representation learning. This shift towards generative self-supervised learning methods not only demonstrates their ability to capture comprehensive data representations and improve performance in NLP and computer vision but also highlights their significant potential in advancing point cloud processing and analysis. Building upon these advancements, our work extends the principles of self-supervised learning from NLP and computer vision to 3D point cloud analysis. By adopting strategies akin to masked token prediction in BERT and reconstruction in Masked Autoencoders, we introduce an attention-driven dynamic masking approach that encourages the model to capture comprehensive structural information from point clouds.

**Self-Supervised Learning for Point Cloud.** Various methods have been investigated for self-supervised representation learning on point clouds (Wang et al., 2024; Liu et al., 2024; Wu et al., 2024; Zhang et al., 2023; 2024; Han et al., 2024). Many previous works focused on generative modeling with generative adversarial networks and autoencoders, aiming to reconstruct input point clouds using different architectural designs (Min et al., 2022; Yu et al., 2022; Sauder & Sievers, 2019; Li et al., 2018a; Achlioptas et al., 2018; Wang et al., 2022). PointMAE (Pang et al., 2022) proposes a effective scheme of masked autoencoders for point cloud self-supervised learning. Point-M2AE (Zhang et al., 2022a) further employs a hierarchical transformer architecture and implements a specific masking strategy. PointGPT (Chen et al., 2024) propose a point cloud auto-regressive generation task to pre-train transformer models. Moreover, contrastive methods also have been extensively explored (Qian et al., 2022; Xue et al., 2023; 2024; Navaneet et al., 2020; Zhang et al., 2021; Xie et al., 2020; Huang et al., 2023). DepthContrast (Zhang et al., 2021) generates augmented depth maps and conducts instance discrimination on the extracted global features. MVIF (Jing et al., 2020) employs cross-modal and cross-view invariance constraints to enable self-supervised learning of modal- and view-invariant features. OcCo (Wang et al., 2021) aims to reconstruct the original point cloud from an occluded version observed in camera views. Some studies focus on integrating cross-modal information, utilizing knowledge from language or image models to enhance 3D learning (Qi et al., 2023; Dong et al., 2022; Qi et al., 2024; Saito & Poovvancheri, 2024). PointCLIP (Zhang et al., 2022b) facilitates the alignment between point clouds encoded by CLIP and corresponding 3D category text descriptions, enhancing cross-modal understanding. PointCLIP V2 (Zhu et al., 2023a) uses a shape projection module to guide CLIP in generating more realistic depth maps and prompts a GPT model to create 3D-specific text for CLIP's textual encoder input. Unlike previous approaches that primarily rely on random or fixed masking strategies in generative frameworks, PointACL leverages the model's inherent attention distribution to dynamically select high-attention regions for masking. This encourages the model to focus on under-represented low-attention areas, enabling it to learn more comprehensive and robust point cloud features.

## 3 METHODS

The overall framework of PointACL is illustrated in Figure 2. First, the Attention-driven Dynamic Masking module generates an attention-guided masked point cloud. Both the masked point cloud and the original input point cloud are then fed into the shared backbone model to obtain the global features of each input. By aligning the features from these two branches with contrastive loss, we guide the model to focus on the low-attention regions of the point clouds, thereby improving feature discrimination and generalization. During the pre-training stage, we train the model using a combination of contrastive loss and the original pre-training loss—such as the reconstruction loss from PointMAE (Pang et al., 2022) or the generation loss from PointGPT (Chen et al., 2024). After

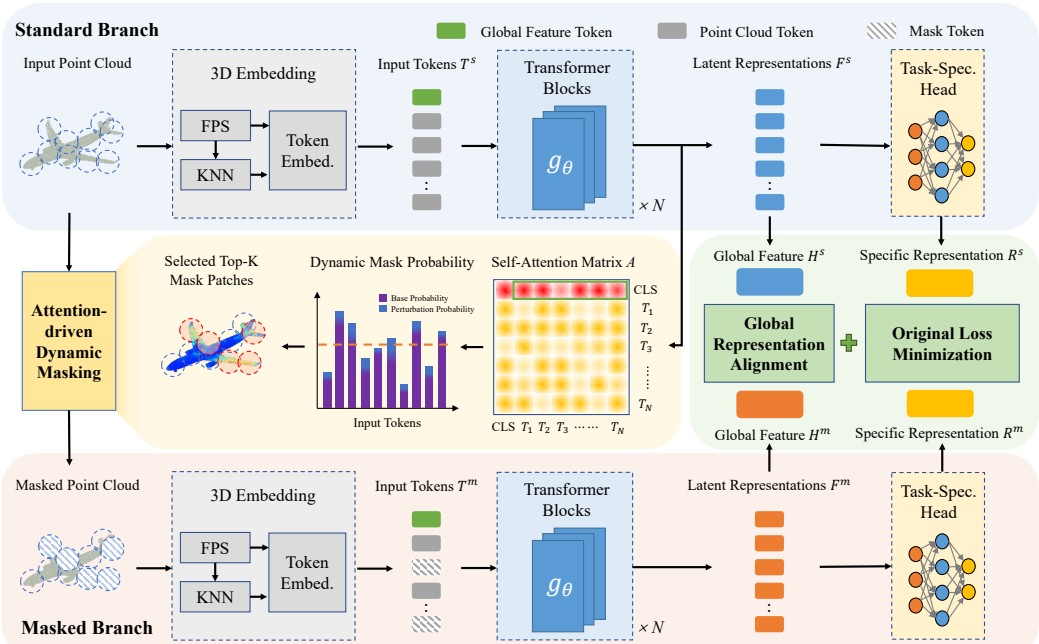

Figure 2: **Overview of the PointACL Framework.** PointACL consists of two branches that share the same weights: a standard mode branch and a masked mode branch. An attention-driven dynamic masking module generates a masked point cloud by selecting less activated patches from the output of the standard mode branch. Both branches process their respective inputs through the shared Transformer blocks to obtain latent representations. Finally, a joint contrastive loss is used to align the representations of these two branches.

pre-training, we employ the backbone model without the masking strategy, leveraging the learned latent representations for downstream tasks.

### 3.1 PRELIMINARY

**Transformer-based self-supervised learning.** Given a point cloud $X \in \mathbb{R}^{P \times 3}$, we utilize Farthest Point Sampling (FPS) and K-Nearest Neighbors (KNN) algorithms to identify $n$ center points $C$ and their corresponding $k$ nearest neighbors, forming $n$ point patches $P$. Following the previous methods (Pang et al., 2022; Chen et al., 2024), each point patch is normalized to integrate local information. A lightweight token embedding module, implemented via PointNet, subsequently transforms these normalized local patches into trainable point tokens $T$. These point tokens, together with positional embeddings, are input into the transformer blocks to produce latent representations $F$. For different tasks, these latent representations are input into task-specific heads, where they are transformed into specific representations adapted to the task. The learning pipeline based on the Transformer architecture is as follows:

$$F = Transformer(T), \tag{1}$$

$$R = Head_{Task-Spec.}(F). \tag{2}$$

For Point-MAE, $Head_{Task-Spec.}$ denotes the reconstruction head. For PointGPT, $Head_{Task-Spec.}$ denotes the prediction head.

**Point patch attention.** We employ the self-attention mechanism in the transformer architecture to compute the attention weights of point patches relative to the global feature. A new set of input tokens $T \in \mathbb{R}^{(N+1) \times d}$, consisting of the point tokens $T^p \in \mathbb{R}^{N \times d}$ and a learnable global feature token $T^f \in \mathbb{R}^{1 \times d}$, is utilized to compute the queries $Q \in \mathbb{R}^{(N+1) \times d}$, keys $K \in \mathbb{R}^{(N+1) \times d}$, and values $V \in \mathbb{R}^{(N+1) \times d}$. The attention matrix $A$ is subsequently derived from the dot product of the queries and keys. Since the first element of the input tokens $T_1$ corresponds to the global feature token, the first row of the attention matrix can be interpreted as the contribution of each token to the

global feature. Considering the output tokens depend on both the attention matrix and the values, we incorporate the norm of $V_j$ when determining the significance score of token $j$. Consequently, the attention matrix and significance score for point patch $j$ are computed as follows:

$$A = Softmax(QK^T/\sqrt{d}), \tag{3}$$

$$S_j = \frac{A_{1,j} \times \|V_j\|}{\sum_{i=2} A_{1,i} \times \|V_i\|}, \tag{4}$$

where $i, j \in 2, ..., N + 1$. For a multi-head attention layer, we compute the significance scores for each head separately and aggregate them by taking the sum over all heads.

### 3.2 ATTENTION-DRIVEN DYNAMIC MASKING

To fully harness the advantages of the self-attention mechanism and mitigate the model's reliance on a small subset of key patches, we propose an attention-driven dynamic masking strategy, which guides the model to focus on low-attention regions and enforces a more comprehensive understanding of the global structure in challenging scenarios by dynamically masking high-attention areas.

A straightforward idea is to mask the top $k$ patches with the highest significance scores, as they are key to the model's understanding of the point cloud. However, a fixed masking probability merely shifts the model's attention without engaging a broader set of patches. As the model becomes reliant on new areas of focus, it similarly falls into the trap of limited comprehension of the point cloud. Our primary objective is to ensure that high-attention regions have a higher likelihood of being masked. Therefore, we suggest a dynamic masking. Specifically, we construct an updatable base masking probability using the latest self-attention significance scores, prioritizing the masking of patches that currently contribute significantly to the global features. Additionally, a perturbation probability, derived from a uniform distribution $U[0, 1]$, is introduced to enhance the variability of the masking probability. Based on this concept, the final dynamic masking probability $p_{dy}$ is expressed as:

$$p_{dy} = \log\left(Softmax(S/\tau_{pro})\right) - \log\left(-\log\varepsilon\right), \quad \varepsilon \in U[0, 1], \tag{5}$$

where $\tau_{pro}$ is a temperature hyperparameter which controls the sharpness of the base masking probability. A lower temperature (less than 1) results in a sharper distribution, meaning that regzions with the highest attention are more likely to be masked. Based on the dynamic masking probability, we apply simple Top-K strategy to select the $k$ point patches $P^{mask} \in \mathbb{R}^{K \times 3}$ to be masked:

$$P^{mask} = \text{Top-K}(p_{dy}, k). \tag{6}$$

They are then replaced with learnable mask tokens. In this manner, regions that attract high attention are more likely to be masked, promoting a deeper understanding of the global structure by the model.

### 3.3 LEARNING OBJECTIVE

To further improve the model's feature discrimination and generalization, we introduce the contrastive loss to the pre-training stage, which combines the original pre-training loss with a contrastive learning objective, enabling the model to retain task-specific learning capabilities while enhancing its global understanding.

**Global representation alignment.** The dynamically selected masked token $T^m$ and the standard token $T^s$ are both input into a shared-weight model, producing two distinct levels of point cloud latent representations $F^m$ and $F^s$. Unlike the masked latent representations, the complete point cloud retains all original information. Although the masking strategy results in the loss of some regional details, both representations still correspond to the same underlying point cloud entity. Therefore, we expect the global features extracted from the masked point cloud to align with those derived from the standard point cloud. This alignment ensures that the model captures the overall structure of the point cloud without over-relying on specific local regions. To achieve this, we introduce a contrastive learning objective:

$$\mathcal{L}_{contra} = -\frac{1}{2b} \sum_i \left( \log \frac{\exp(H_i^m \cdot H_i^s/\tau_{sim})}{\sum_j \exp(H_i^m \cdot H_j^s/\tau_{sim})} + \log \frac{\exp(H_i^s \cdot H_i^m/\tau_{sim})}{\sum_j \exp(H_i^s \cdot H_j^m/\tau_{sim})} \right), \tag{7}$$

where $b$ is the number of point clouds in a batch; $\tau_{sim}$ is a temperature hyperparameter; $H_i^m$ and $H_i^s$ are the normalized projection features of $F_i^m$ and $F_i^s$. By omitting the high-attention regions in the masked point clouds, the contrastive objective incentivizes the model to focus on and extract valuable information from less emphasized areas. This process facilitates the learning of a more holistic latent representation, thereby improving the model's capacity to effectively differentiate between various point cloud objects.

**Contrastive learning enhancement.** While traditional contrastive learning methods have demonstrated significant success in unsupervised and self-supervised learning, relying solely on contrastive loss may weaken the model's performance on specific tasks. This limitation arises from the model's inability to fully exploit the advantages of the existing framework. To address this issue, we propose that a better solution is to integrate the contrastive loss into the existing framework. This approach preserves the model's task-specific learning capabilities while leveraging contrastive learning to further improve its global understanding and generalization capacity. The proposed total loss is formulated as follows:

$$\mathcal{L}_{total} = \mathcal{L}_{origin} + \lambda \mathcal{L}_{contra}, \tag{8}$$

where $\mathcal{L}_{origin}$ represents the original loss in the existing framework; $\lambda$ is a weight hyperparameter that controls the contribution of contrastive learning loss. During the pre-training phase, Point-MAE's original pre-training loss $\mathcal{L}_{origin}$ is equivalent to the reconstruction loss $\mathcal{L}_{re}$. For PointGPT, $\mathcal{L}_{origin}$ refers to the generation loss $\mathcal{L}_{ge}$. Therefore, we jointly optimizes the reconstruction (or generation) and contrastive losses, ensuring that the model not only achieves high-quality reconstructions (or generations) but also learns globally consistent feature representations. Through this strategy, PointACL exhibits strong potential for adaptability and scalability across a wide range of multi-task learning scenarios, ultimately improving the model's overall performance.

## 4 EXPERIMENTS

### 4.1 EXPERIMENTAL SETUP

**Datasets.** We evaluate PointACL framework on three benchmark datasets commonly used in 3D point cloud analysis. ***ScanObjectNN*** (Uy et al., 2019) comprises approximately 15,000 real-world 3D objects from 15 categories derived from indoor RGB-D scans, presenting challenges like background clutter, occlusions, and sensor noise, thus testing the robustness and generalization of our method in realistic scenarios. ***ModelNet40*** (Wu et al., 2015) is a synthetic dataset with 12,311 CAD models across 40 categories, split into 9,843 for training and 2,468 for testing, providing clean and uniformly sampled point clouds ideal for assessing classification performance without real-world complexities. ***ShapeNetPart*** (Yi et al., 2016) contains 16,881 models across 16 categories, each annotated with point-level part labels totaling 50 classes, enabling evaluation of fine-grained part segmentation and demonstrating the versatility of our approach in detailed 3D understanding tasks.

**Backbone models.** To evaluate the seamless integration of the proposed method into existing Transformer-based models for point cloud processing, we employed different backbone architectures, specifically Point-MAE and PointGPT-S, to validate the algorithm's effectiveness. Experimental results across various tasks indicate that the method is adaptable and enhances the performance of these Transformer architectures, thereby demonstrating its versatility and practical applicability.

**Experimental details.** Our input point clouds are obtained by sampling 1,024 points from each raw point cloud. Each point cloud is then divided into 64 patches with 32 points each. The PointACL model is pre-trained for a total of 600 epochs: the first 300 epochs focus on the original task alone, and the next 300 epochs incorporate both original pre-training and contrastive learning objectives. We use the Adam optimizer with an initial learning rate of 0.001, a weight decay of 0.05, and a batch size of 128. The learning rate is adjusted using a cosine decay schedule. All experiments are implemented using the PyTorch framework and conducted on four NVIDIA V100 GPUs.

### 4.2 EXPERIMENTAL RESULTS

**Real-world object classification on ScanObjectNN.** Table 1 compares our proposed PointACL method with existing approaches on the ScanObjectNN dataset across OBJ-BG, OBJ-ONLY, and PB-T50-RS settings. Our PointACL consistently outperforms these state-of-the-art methods. Com-

Table 1: **Object classification on ScanObjectNN and ModelNet40.** We report the Top-1 classification accuracy (%) of PointACL with Point-MAE and PointGPT-S as backbones respectively. On ScanObjectNN, * denotes using simple rotational augmentation for training. On ModelNet40, * denotes the results obtained by voting.

| Methods | Reference | ScanObjectNN | | | ModelNet40 |
| --- | --- | --- | --- | --- | --- |
| | | OBJ-BG | OBJ-ONLY | PB-T50-RS | |
| *Supervised Learning Only* | | | | | |
| PointNet (Qi et al., 2017a) | CVPR 17 | 73.3 | 79.2 | 68.0 | 89.0 |
| PointNet++ (Qi et al., 2017b) | NeurIPS 17 | 82.3 | 84.3 | 77.9 | 90.2 |
| PointCNN (Li et al., 2018b) | NeurIPS 18 | 86.1 | 85.5 | 78.5 | 91.7 |
| DGCNN (Wang et al., 2019) | TOG 19 | 82.8 | 86.2 | 78.1 | 92.0 |
| PRANet (Cheng et al., 2021) | TIP 21 | - | - | 81.0 | 92.9 |
| MVTN (Hamdi et al., 2021) | ICCV 21 | - | - | 82.8 | 93.8 |
| PointNeXt (Qian et al., 2022) | NeurIPS 22 | - | - | 87.7 | 92.9 |
| PointMLP (Ma et al., 2022) | ICLR 22 | - | - | 85.4 | 94.1 |
| RepSurf-U (Ran et al., 2022) | CVPR 22 | - | - | 84.3 | 93.8 |
| ADS (Hong et al., 2023) | ICCV 23 | - | - | 87.5 | 94.0 |
| *with Self-Supervised Representation Learning* | | | | | |
| Point-BERT (Yu et al., 2022) | CVPR 22 | 87.4 | 88.1 | 83.1 | 92.7 |
| MaskPoint (Liu et al., 2022) | CVPR 22 | 89.3 | 88.1 | 84.3 | 92.6 |
| Point-M2AE (Zhang et al., 2022a) | NeurIPS 22 | 91.2 | 88.8 | 86.4 | 93.4 |
| PointDif (Zheng et al., 2024) | CVPR 24 | 93.3 | 91.9 | 87.6 | - |
| GPM (Li et al., 2024) | CVPR 24 | 90.2 | 90.0 | 84.8 | 93.3 |
| Point-MAE (Pang et al., 2022) | ECCV 22 | 90.0 | 88.3 | 85.2 | 93.2 |
| **+PointACL** | - | 90.9 | 88.8 | 85.4 | 93.7 |
| ↑ *Improve* | - | +0.9 | +0.5 | +0.2 | +0.5 |
| PointGPT-S (Chen et al., 2024) | NeurIPS 23 | 91.6 | 90.0 | 86.9 | 93.3 |
| **+PointACL** | - | 92.3 | 91.6 | 87.1 | 93.5 |
| ↑ *Improve* | - | +0.7 | +1.6 | +0.2 | +0.2 |
| Point-MAE* (Pang et al., 2022) | ECCV 22 | 92.8 | 91.2 | 89.0 | 93.8 |
| **+PointACL*** | - | 93.1 | 91.7 | 89.2 | **94.1** |
| ↑ *Improve* | - | +0.3 | +0.5 | +0.2 | +0.3 |
| PointGPT-S* (Chen et al., 2024) | NeurIPS 23 | 93.4 | 92.4 | 89.2 | 94.0 |
| **+PointACL*** | - | **94.5** | **93.5** | **89.9** | **94.1** |
| ↑ *Improve* | - | +1.1 | +1.1 | +0.7 | +0.1 |

pared to Point-MAE (Pang et al., 2022), PointACL achieves higher accuracies by +0.9%, +0.5%, and +0.2% on OBJ-BG, OBJ-ONLY, and PB-T50-RS, respectively. Against PointGPT-S (Chen et al., 2024), PointACL attains improvements of +0.7%, +1.6%, and +0.2% on the same splits. With simple rotational augmentation (marked with *), PointACL sets new state-of-the-art results, achieving up to 94.5% on OBJ-BG, 93.5% on OBJ-ONLY and 89.9% on PB-T50-RS. These results demonstrate that PointACL effectively enhances feature representation for point cloud data, particularly in challenging scenarios with background noise and object perturbations. The consistent performance gains across all settings highlight the robustness and efficacy of our approach.

**Synthetic object classification on ModelNet40.** Table 1 presents the performance of our proposed PointACL method compared to existing self-supervised learning approaches on the ModelNet40 dataset, evaluated both without voting and with voting. Our PointACL achieves an accuracy of 93.7% without voting and 94.1% with voting, surpassing previous methods without adding additional parameters. Specifically, compared to Point-MAE, PointACL improves accuracy by +0.5% without voting and +0.3% with voting. When compared to PointGPT-S, our method achieves gains of +0.2% and +0.1%, respectively. These results demonstrate that PointACL effectively enhances feature representation learning for 3D point cloud data, leading to superior classification performance on ModelNet40.

**Few-shot classification on ModelNet40.** Our PointACL framework was evaluated on the ModelNet40 dataset under few-shot learning settings, and the results are presented in Table 2. Following standard practice, we carry out 10 separate experiments for each setting and reported mean accuracy

Table 2: **Few-shot classification on ModelNet40.** We report the mean accuracy (%) with standard deviation over 10 independent experiments.

| Methods | 5-way | | 10-way | |
|---|---|---|---|---|
| | 10-shot | 20-shot | 10-shot | 20-shot |
| *Supervised Learning Only* | | | | |
| PointNet | 52.0±3.8 | 57.8±4.9 | 46.6±4.3 | 35.2±4.8 |
| PointNet-CrossPoint | 90.9±1.9 | 93.5±4.4 | 84.6±4.7 | 90.2±2.2 |
| DGCNN | 31.6±2.8 | 40.8±4.6 | 19.9±2.1 | 16.9±1.5 |
| DGCNN-CrossPoint | 92.5±3.0 | 94.9±2.1 | 83.6±5.3 | 87.9±4.2 |
| *with Self-Supervised Representation Learning* | | | | |
| Point-BERT | 94.6±3.1 | 96.3±2.7 | 91.0±5.4 | 92.7±5.1 |
| MaskPoint | 95.0±3.7 | 97.2±1.7 | 91.4±4.0 | 93.4±3.5 |
| Point-M2AE | 96.8±1.8 | 98.3±1.4 | 92.3±4.5 | 95.0±3.0 |
| Point-MAE | 96.3±2.5 | 97.8±1.8 | 92.6±4.1 | 95.0±3.0 |
| **+PointACL** | **96.7±2.7** | **98.2±1.6** | **92.8±4.0** | **95.3±3.2** |
| PointGPT | 96.8±2.0 | 98.6±1.1 | 92.6±4.6 | 95.2±3.4 |
| **+PointACL** | **97.1±2.3** | **98.8±1.3** | **93.0±4.0** | **95.6±3.0** |

Table 3: **Part segmentation performance on the ShapeNetPart dataset.** We report the mean Intersection over Union (mIoU) across instances (Ins.) and classes (Cls.).

| Methods | Ins. mIoU | Cls.mIoU |
|---|---|---|
| *Supervised Learning Only* | | |
| PointNet | 83.7 | 80.4 |
| PointNet++ | 85.1 | 81.9 |
| DGCNN | 85.2 | 82.3 |
| *with Self-Supervised Representation Learning* | | |
| Point-BERT | 85.6 | 84.1 |
| GPM | 85.8 | 84.2 |
| Point-MAE | 86.1 | 84.2 |
| **+PointACL** | **86.2** | **85.0** |
| ↑ *Improve* | +0.1 | +0.8 |
| PointGPT-S | 86.2 | 84.1 |
| **+PointACL** | **86.3** | **84.4** |
| ↑ *Improve* | +0.1 | +0.3 |

along with the standard deviation. Compared to both supervised learning methods and other self-supervised representation learning approaches, PointACL consistently achieves higher accuracy. In the 5-way 10-shot task, our method attains an accuracy of 97.1% with a standard deviation of 2.3%, outperforming previous methods. Similarly, in the 10-way 20-shot setting, PointACL achieves an accuracy of 95.6%, demonstrating superior generalization with limited labeled data.

**Part segmentation on ShapeNetPart.** We evaluated the effectiveness of our PointACL framework on the part segmentation task using the ShapeNetPart dataset, as shown in Table 3. PointACL achieves superior performance compared to both traditional supervised models like PointNet and DGCNN and recent self-supervised methods like Point-MAE and PointGPT-S. Specifically, our method attains an instance mIoU of 86.2% and a class mIoU of 85.0%, showing improvements over existing methods. These results demonstrate that our attention-driven contrastive learning strategy effectively enhances the model's ability to segment parts in complex 3D shapes, confirming the efficacy of PointACL in advancing the state-of-the-art in point cloud segmentation.

## 4.3 ABLATION STUDIES

In our ablation studies, we use PointGPT-S as the backbone and conduct extensive experiments on ScanObjectNN to validate the effectiveness of each component. More importantly, we also performed robustness tests to assess the model's resilience under various noisy environments, including Gaussian noise, rotation, scaling, and point dropout.

**Mask strategy and loss optimization function.** Table 4(a) summarizes the ablation study on different mask strategies and loss functions for the OBJ-BG and OBJ-ONLY settings. We evaluated *No Mask*, *Random Mask*, *Low-Attention Mask*, and *High-Attention Mask* strategies, combined with the original generation loss ($\mathcal{L}_{origin}$) and the proposed contrastive loss ($\mathcal{L}_{contra}$). Without masking, the baseline model achieves accuracies of 91.6% (OBJ-BG) and 90.0% (OBJ-ONLY). Applying a *Random Mask* slightly improves performance, and adding $\mathcal{L}_{contra}$ further enhances accuracies to 92.1% and 90.9%. The *Low-Attention Mask* strategy yields marginal gains, but when combined with $\mathcal{L}_{contra}$, it reaches 92.0% and 91.4%. The *High-Attention Mask* strategy delivers the best results. With $\mathcal{L}_{origin}$ alone, it attains 91.9% (OBJ-BG) and 91.2% (OBJ-ONLY). Incorporating $\mathcal{L}_{contra}$ boosts performance to 92.3% and 91.6%, the highest in our study. This demonstrates that masking the most informative regions forces the model to learn robust features from less informative areas, and the contrastive loss $\mathcal{L}_{contra}$ enhances feature discrimination. In summary, the combination of the High-Attention Mask strategy and the contrastive loss $\mathcal{L}_{contra}$ significantly improves classification accuracy, highlighting the effectiveness of both components in our method.

**Mask ratio.** As shown in Table 4(b), the model's performance improves with increasing masking, peaking at a mask ratio of $\mathcal{R} = 0.6$. which achieves classification accuracies of 92.3% on the OBJ-BG dataset and 91.6% on the OBJ-ONLY dataset. However, a higher mask ratio (0.8) hinders the

Table 4: **Ablation studies of components in PointACL.** We report the overall accuracy (%) on ScanObjectNN with PointGPT-S as our backbone. The settings adopted by PointACL are  marked .

(a) Mask Strategy and Loss Optimization Function.

| Mask Strategy | $\mathcal{L}_{origin}$ | $\mathcal{L}_{contra}$ | OBJ-BG | OBJ-ONLY |
|---|---|---|---|---|
| NO Mask | ✓ | - | 91.6 | 90.0 |
| Random Mask | ✓ | - | 91.7 | 90.5 |
| Random Mask | ✓ | ✓ | 92.1 | 90.9 |
| Low-Attention Mask | ✓ | - | 91.7 | 90.7 |
| Low-Attention Mask | ✓ | ✓ | 92.0 | 91.4 |
| High-Attention Mask | ✓ | - | 91.9 | 91.2 |
| **High-Attention Mask** | ✓ | ✓ | **92.3** | **91.6** |

(b) Mask Ratio.

| $\mathcal{R}$ | OBJ-BG | OBJ-ONLY |
|---|---|---|
| 0.2 | 91.7 | 90.9 |
| 0.4 | 91.9 | 91.2 |
| **0.6** | **92.3** | **91.6** |
| 0.8 | 91.6 | 90.5 |

(c) Probability Temperature.

| $\tau_{pro}$ | OBJ-BG | OBJ-ONLY |
|---|---|---|
| 0.3 | 91.6 | 91.4 |
| **0.5** | **92.3** | **91.6** |
| 0.7 | 92.1 | 91.6 |
| 0.9 | 91.9 | 91.4 |

(d) Contrastive Loss Weight.

| $\lambda$ | OBJ-BG | OBJ-ONLY |
|---|---|---|
| 0.4 | 91.7 | 90.9 |
| **0.6** | **92.3** | **91.6** |
| 0.8 | 92.1 | 90.9 |
| 1 | 91.7 | 91.0 |

Table 5: **Robustness analysis.** We report the classification accuracy (%) with four noisy environments: Gaussian noise, rotation, scaling, and droppoint on ScanObjectNN.

| DataSet | Methods | Gaussian Noise | | Rotation | | | Scaling | DropPoint | |
|---|---|---|---|---|---|---|---|---|---|
| | | $\sigma$=0.01 | $\sigma$=0.03 | X[-30 30] | Y[-30 30] | Z[-30 30] | (0.5, 1.5) | 0.2 | 0.6 |
| OBJ-BG | Point-MAE | 77.5 | 47.2 | 72.1 | 87.6 | 72.5 | 86.2 | 87.4 | 84.9 |
| | **+PointACL** | 81.8 | 60.6 | 77.3 | 90.5 | 77.3 | 89.3 | 90.7 | 89.7 |
| | ↑ *Improve* | +4.3 | +13.4 | +5.2 | +2.9 | +4.8 | +3.1 | +3.3 | +4.8 |
| | PointGPT-S | 78.6 | 51.5 | 72.3 | 89.3 | 74.0 | 88.3 | 90.7 | 85.0 |
| | **+PointACL** | 81.8 | 57.8 | 76.8 | 91.9 | 79.2 | 90.4 | 91.4 | 86.1 |
| | ↑ *Improve* | +3.2 | +6.3 | +4.5 | +2.6 | +5.2 | +2.1 | +0.7 | +1.1 |
| OBJ-ONLY | Point-MAE | 70.9 | 37.0 | 75.4 | 86.7 | 74.9 | 84.0 | 86.6 | 84.5 |
| | **+PointACL** | 76.2 | 54.2 | 78.5 | 88.6 | 79.7 | 86.7 | 88.5 | 87.6 |
| | ↑ *Improve* | +5.3 | +17.2 | +3.1 | +1.9 | +4.8 | +2.7 | +1.9 | +3.1 |
| | PointGPT-S | 71.2 | 39.4 | 72.3 | 89.3 | 74.5 | 86.6 | 89.7 | 85.9 |
| | **+PointACL** | 73.3 | 41.3 | 79.9 | 92.3 | 81.8 | 90.0 | 91.2 | 87.4 |
| | ↑ *Improve* | +2.1 | +1.9 | +7.6 | +3.0 | +7.3 | +3.4 | +1.5 | +1.5 |

model's performance due to the loss of critical information necessary for accurate predictions. This emphasizes the importance of an optimal mask ratio that balances data complexity with sufficient information retention for robust classification.

**Probability temperature.** We further explore the effects of varying the temperature hyperparameter in the dynamic masking probability. Results in Table 4(c) indicate that setting $\tau_{pro}$ to 0.5 yields the highest classification accuracies, achieving 92.3% on OBJ-BG and 91.6% on OBJ-ONLY. This suggests that this temperature value effectively masks the region of higher attention while maintaining a certain level of dynamic selection, allowing the model to improve global understanding.

**Contrastive loss weight.** The analysis of contrastive loss weight in Table 4(d) demonstrates that $\lambda = 0.6$ strikes the best balance between the original loss and the contrastive loss. This optimal balance maximizes overall performance and enhances accuracy across both datasets. By fine-tuning the loss weights, PointACL effectively leverages contrastive learning to improve global understanding and generalization capabilities while maintaining task-specific performance.

**Robustness analysis.** To assess the robustness of our PointACL framework, we conducted experiments on the ScanObjectNN dataset under different noisy environments, including Gaussian noise, rotation, scaling, and point dropout, as detailed in Table 5. Compared to the state-of-the-art models Point-MAE and PointGPT-S, our method consistently achieves higher classification accuracies across both OBJ-BG and OBJ-ONLY settings. For instance, under Gaussian noise with

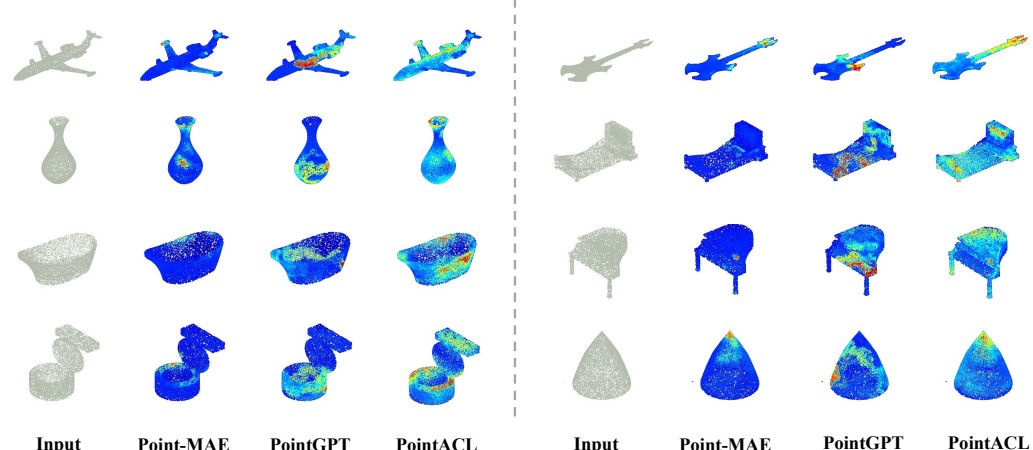

| Input | Point-MAE | PointGPT | PointACL | Input | Point-MAE | PointGPT | PointACL |

Figure 3: **Attention visualization of PointACL with Point-MAE and PointGPT.** Patches with high attention are closer to red, while patches with low attention are closer to blue. Point-MAE is employed as the backbone of our proposed PointACL.

$\sigma = 0.03$, PointACL outperforms Point-MAE by up to 13.4% and PointGPT-S by 6.3%. Similar improvements are observed with rotational perturbations around the X, Y, and Z axes, scaling factors ranging from 0.5 to 1.5, and point dropout rates of 20% and 60%. These results demonstrate that our attention-driven dynamic masking strategy and contrastive learning significantly enhance the model's resilience to noise and transformations. The consistent performance gains highlight PointACL's ability to capture more comprehensive and discriminative features, making it robust in real-world scenarios where point clouds often contain noise, occlusions, and varying orientations.

### 4.4 QUALITATIVE ANALYSIS

As shown in Figure 3, we visualize the classification heatmaps generated by different models (Point-MAE, PointGPT, and our proposed PointACL), which reveals significant distinctions in how each model attends to various regions of the point clouds. PointACL exhibits a more balanced and comprehensive activation across both prominent and under-represented areas of the input data. This observation directly corresponds with the issues highlighted in our introduction, where we identified that existing Transformer-based models tend to overlook latent information in less prominent regions, resulting in limited global understanding and increased sensitivity to perturbations. By integrating our attention-driven dynamic masking strategy, PointACL effectively encourages the model to focus on under-attended regions, thus enhancing its ability to capture the global structural information of the point cloud. Additionally, the contrastive learning further refines feature discrimination and generalization. In contrast, the heatmaps of Point-MAE and PointGPT indicate a predominant focus on high-attention regions, potentially neglecting valuable information elsewhere. The richer and more evenly distributed activations in PointACL's heatmaps substantiate its superior capacity for comprehensive point cloud analysis, confirming the efficacy of our approach in addressing the limitations of existing models and underscoring the advantages of our methods.

## 5 CONCLUSION

In this work, we present PointACL, an attention-driven contrastive learning framework. By integrating an attention-driven dynamic masking strategy with contrastive learning, our method leverages the model's inherent attention distribution to dynamically mask high-attention regions. This approach guides the network to focus on under-attended low-attention areas, enabling it to learn more comprehensive and robust point cloud feature representations. Our extensive experiments demonstrate that PointACL significantly enhances the understanding of global structures in point clouds, leading to notable improvements across various tasks, including object classification, part segmentation, and few-shot learning. We hope that our work can inspire more explorations of self-supervised learning and contrastive learning in point cloud understanding.

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

# A APPENDIX

## A.1 ABLATION STUDY ANALYSIS

**Training strategy analysis.** Given that PointACL determines mask patches based on the attention weights of the backbone network, we suggest two strategies for obtaining these attention weights. The first strategy initializes the network with random attention and applies the attention-driven dynamic masking for adaptive attention refinement during subsequent training. Following standard protocol, the model undergoes pre-training for 300 epochs. This approach does not incur any additional training overhead. The second strategy, by contrast, employs attention learned from the standard branch for initialization, aiming to dynamically adjust the model's dependencies in a targeted manner. This method necessitates 300 epochs of pre-training in the standard branch, followed by another 300 epochs in the dual branch, resulting in a total of 600 epochs.

Furthermore, we introduce PointACL during the fine-tuning phase of downstream tasks to further evaluate the scalability and effectiveness of our approach. Two strategies are employed here as well: one leverages the pre-trained attention for initialization, while the other requires an additional 300 epochs of training to obtain attention learned from the standard branch for initialization.

Our experimental results, presented in Table 6, demonstrate the inherent advantages of our PointACL over existing approaches (such as Point-MAE and PointGPT-S) under the same training time and training phase. For Point-MAE, with 300 pre-training epochs or 300 fine-tuning epochs, PointACL achieves an accuracy of 90.5% on the OBJ-BG dataset, surpassing Point-MAE's 90.0% by a margin of 0.5%. This improvement persists when both methods are trained for 600 epochs during the fine-tuning phase, with PointACL reaching 90.9% accuracy compared to Point-MAE's 90.0%. Similarly, when evaluating against PointGPT-S, PointACL continues to exhibit superior performance. With both models trained for 300 fine-tuning epochs on OBJ-ONLY, PointACL attains an accuracy of 90.9% compared to PointGPT-S's 90.2%. Even when the training epochs are extended to 600, PointACL maintains its advantage, achieving 91.6% accuracy, outperforming PointGPT-S by 1.4%. On the OBJ-BG dataset, a similar pattern is observed, where PointACL consistently outperforms PointGPT-S regardless of training duration.

The superior performance of PointACL across various datasets, training epochs, and application phases validates the efficacy of our framework. It demonstrates the performance gains of PointACL are not a consequence of longer training times but are a direct result of designed framework contributions—namely, the attention-driven dynamic masking strategy with contrastive learning. By focusing on under-attended regions and enhancing feature discrimination, PointACL effectively captures both global and local features, leading to enhanced robustness and generalization.

Table 6: **Training strategy analysis.** We report the classification accuracy (%) on ScanObjectNN.

| DataSet | Methods | Pre-Training Epoch | | Finetune Epoch | |
|---|---|---|---|---|---|
| | | 300 | 600 | 300 | 600 |
| OBJ-BG | Point-MAE | 90.0 | 90.2 | 90.0 | 90.0 |
| | **+PointACL** | 90.5 | 90.5 | 90.5 | 90.9 |
| | ↑ *Improve* | +0.5 | +0.3 | +0.5 | +0.9 |
| | PointGPT-S | 91.6 | 91.7 | 91.6 | 91.9 |
| | **+PointACL** | 91.9 | 91.9 | 92.1 | 92.3 |
| | ↑ *Improve* | +0.3 | +0.2 | +0.5 | +0.4 |
| OBJ-ONLY | Point-MAE | 88.3 | 88.5 | 88.3 | 88.3 |
| | **+PointACL** | 88.8 | 89.8 | 89.2 | 88.8 |
| | ↑ *Improve* | +0.5 | +1.3 | +0.9 | +0.5 |
| | PointGPT-S | 90.0 | 90.2 | 90.0 | 90.2 |
| | **+PointACL** | 90.5 | 91.4 | 90.9 | 91.6 |
| | ↑ *Improve* | +0.5 | +1.2 | +0.9 | +1.4 |

**Mask strategy analysis.** We conduct a thorough investigation into the effects of various masking strategies and masking ratios on the classification performance under the OBJ-BG and OBJ-ONLY

Table 7: **Mask strategy analysis.** We report the classification accuracy (%) on ScanObjectNN.

| Mask Strategy | Mask Ratio | Mask Probability | OBJ-BG | OBJ-ONLY |
|---|---|---|---|---|
| Random Mask | 0.2 | - | 91.6 | 90.0 |
| | 0.4 | - | 91.7 | 90.2 |
| | 0.6 | - | 92.1 | 90.9 |
| | 0.8 | - | 90.5 | 89.8 |
| Low-Attention Mask | 0.2 | Fixed | 91.6 | 90.0 |
| | 0.4 | Fixed | 91.9 | 90.2 |
| | 0.6 | Fixed | 91.7 | 90.5 |
| | 0.8 | Fixed | 91.1 | 90.0 |
| High-Attention Mask | 0.2 | Fixed | 91.9 | 90.2 |
| | 0.4 | Fixed | 91.9 | 90.9 |
| | 0.6 | Fixed | 91.9 | 90.9 |
| | 0.8 | Fixed | 91.3 | 90.0 |
| High-Attention Mask | 0.2 | Dynamic | 91.7 | 90.9 |
| | 0.4 | Dynamic | 91.9 | 91.2 |
| | 0.6 | Dynamic | **92.3** | **91.6** |
| | 0.8 | Dynamic | 91.6 | 90.5 |

settings. Four distinct masking strategies are evaluated: Random Masking, Low-Attention Masking, High-Attention Masking with Fixed Masking Probability, and High-Attention Masking with Dynamic Masking Probability. The detailed experimental results are presented in Table 7. For the Random Mask strategy, we observe that increasing the mask ratio from 0.2 to 0.6 leads to improved performance, with accuracies peaking at 92.1% on OBJ-BG and 90.9% on OBJ-ONLY when the mask ratio is 0.6. However, further increasing the mask ratio to 0.8 results in a decmidrule in accuracy. This suggests that masking too many patches hinders the model's ability to learn effective representations. The Low-Attention Mask strategy shows a similar trend but does not surpass the performance of the Random Mask. The highest accuracy achieved with this strategy is 91.9% on OBJ-BG at a mask ratio of 0.4, indicating that masking low-attention regions with a fixed probability offers limited benefits in enhancing model performance. When employing the High-Attention Mask with Fixed Mask Probability, the model achieves comparable results to the Random Mask strategy, with a maximum accuracy of 91.9% on OBJ-BG across multiple mask ratios. This suggests that while masking high-attention regions can encourage the model to focus on under-represented areas, a fixed mask probability may not fully capitalize on this strategy's potential.

In contrast, the High-Attention Mask with Dynamic Mask Probability demonstrates notable performance improvements. Specifically, at a mask ratio of 0.6, our model attains the highest accuracies of 92.3% on OBJ-BG and 91.6% on OBJ-ONLY, outperforming all other masking strategies. The dynamic adjustment of the mask probability based on attention weights allows the model to more effectively target and mask the most prominent regions, thereby compelling it to learn richer features from less attended areas. This dynamic approach enhances the model's ability to capture global structural information and reduces its reliance on a limited set of salient features.

The experimental results confirm the effectiveness of the proposed attention-driven dynamic masking strategy, which enhances feature representation and classification performance by encouraging the model to learn from under-attended regions. This approach addresses the limitations of prior methods that overly focus on prominent local features, improving robustness and generalization in 3D point cloud analysis.

## A.2 ROBUSTNESS ANALYSIS

We evaluate the robustness of our method against existing approaches under Gaussian noise conditions using the OBJ-BG and OBJ-ONLY subsets of the ScanObjectNN dataset. To simulate noisy point clouds, we add Gaussian noise $X \sim \mathcal{N}(0, \sigma^2)$ to all points, incrementally increasing the noise level by varying $\sigma$ from 0 to 0.05 with step size = 0.005.

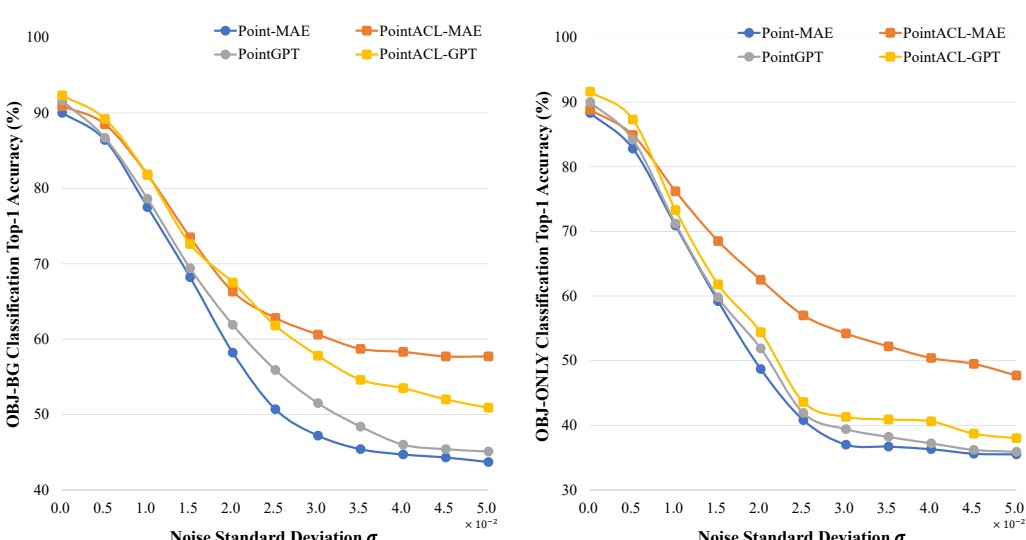

Figure 4: **Gaussian noise analysis on ScanObjectNN.** While the performance of existing methods decmidrules sharply with increasing Gaussian noise, this issue is mitigated by incorporating PointACL. Notably, when Point-MAE is used as the backbone network, our PointACL significantly enhances its robustness, resulting in minimal accuracy degradation.

As illustrated in Figure 4, while the accuracy of all methods decmidrules as the noise standard deviation $\sigma$ increases, PointACL exhibits a slower performance degradation, demonstrating its superior ability to handle noisy point clouds. Notably, PointACL significantly improves the robustness of the Point-MAE backbone and outperforms baseline methods such as Point-MAE and PointGPT, particularly under extreme noise conditions ($\sigma = 0.05$). This improvement can be attributed to our attention-guided dynamic masking strategy, which encourages the model to focus on under-attended regions, thereby enhancing its capacity to capture comprehensive global structural information from point clouds. By not solely relying on salient local features, PointACL mitigates sensitivity to noise-induced perturbations. Additionally, the integration of contrastive learning with the original task further refines feature discrimination, enabling the model to distinguish subtle variations in data even under noisy conditions. The consistently strong performance across both the OBJ-BG and OBJ-ONLY datasets underscores the versatility and reliability of PointACL in diverse settings.

In real-world applications, 3D data is often affected by noise from sensor inaccuracies and environmental factors, making PointACL's robustness to Gaussian noise especially valuable. Its strong performance under such conditions demonstrates its practicality for tasks where data quality is uncertain, underscoring the effectiveness of our framework and its advantage over existing Transformer-based methods.

### A.3 FEATURE DISTRIBUTION ANALYSIS

Figure 5 illustrates the evolution of the global feature distribution using t-SNE during the fine-tuning of PointACL, with Point-MAE as the backbone, on the ModelNet40 dataset. In the early stage feature distribution, the feature space is highly scattered with overlapping clusters, indicating that the backbone has not yet learned to effectively discriminate between different classes. As the backbone starts to align global representations from standard branch and masked branch based on attention-driven dynamic masking, the transitional feature distribution shows a notable improvement, with clusters becoming more distinct. However, there still remains some inter-class overlap.

In the final feature distribution, the clusters are well-separated and compact, reflecting a highly discriminative feature space. The backbone has successfully learned to distinguish between different classes with a high degree of accuracy. The representative clusters at the bottom of each visualization further emphasize this progression, showing a clear transition from mixed and overlapping clusters

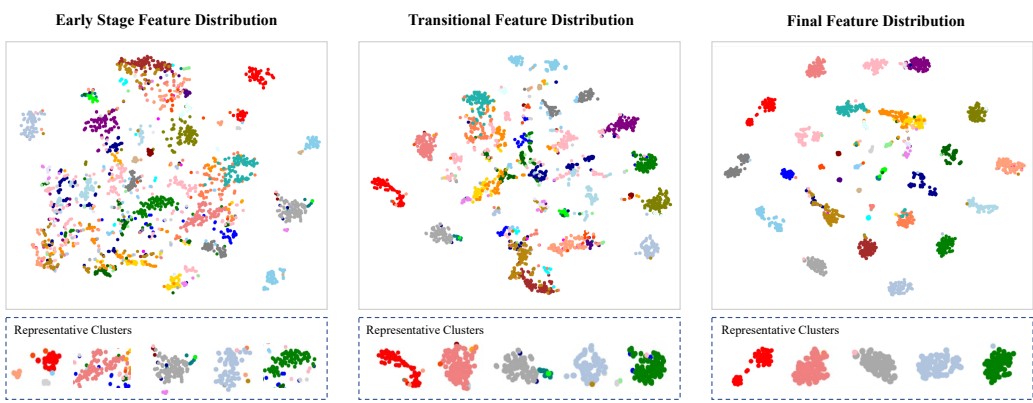

Figure 5: **Feature distribution visualization on ModelNet40. Top:** An overview of the evolution of feature distributions across all 40 classes. **Bottom:** Detailed depiction of the evolution of feature distributions for selected typical classes.

in the early stages to well-defined and isolated clusters in the final stage. These visualizations highlight the effectiveness of the PointACL, demonstrating a clear trajectory of improvement in feature discrimination, culminating in a robust and well-defined feature space.

## A.4   Limatation Analysis

Despite the significant improvements achieved by PointACL, there are still areas that offer opportunities for further enhancement. For example, while our method has been validated on specific datasets, applying it to a broader range of datasets could further demonstrate its generalizability and robustness. Additionally, although we have shown that PointACL integrates seamlessly with certain Transformer-based architectures, exploring its compatibility with an even wider variety of models could highlight its versatility even more. These considerations open avenues for future research to build upon our work and continue advancing the field of point cloud analysis.

## A.5   Future Works

While the proposed PointACL framework has shown significant improvements in point cloud analysis tasks, there are several promising directions for future research to further enhance its capabilities and applications. One potential avenue is the integration of multi-modal data sources to enrich point cloud representations. By incorporating complementary information from modalities such as images, textual descriptions, or LiDAR intensity values, the model can leverage cross-modal correlations to learn more comprehensive and robust feature embeddings. This multi-modal fusion could enhance the model's ability to understand complex scenes and improve performance in tasks like 3D object detection and semantic segmentation. Another direction is the exploration of hierarchical or multi-scale feature learning within the PointACL framework. By capturing features at various spatial resolutions, the model can better represent both local geometric details and global structural contexts. This enhancement could be particularly beneficial for handling large-scale point clouds or scenes with significant variations in point densities. Lastly, applying the PointACL approach to other types of data representations, such as meshes or voxels, could broaden its applicability across different domains in 3D data processing. Exploring transfer learning techniques between these representations may also provide insights into shared structures and features among various 3D data forms.

By pursuing these future research directions, we aim to further advance the capabilities of PointACL, contributing to the development of more robust, efficient, and versatile models for point cloud analysis. These enhancements have the potential to impact a wide range of applications, including robotics, augmented reality, virtual reality, and autonomous navigation, by enabling more accurate and comprehensive understanding of complex 3D environments.

