# OpenReview forum: "PointACL: Point Cloud Understanding via  Attention-Driven Contrastive Learning"
_ICLR.cc/2025/Conference — ICLR 2025 Conference Withdrawn Submission_

### Official Review · Reviewer_F7F3 · 2024-11-02

**Soundness:** 2
**Presentation:** 3
**Contribution:** 2
**Rating:** 5
**Confidence:** 4

**Summary:**

This paper improves previous contrastive learning for point cloud model pretraining. It analyzes the issue in previous works: overlook latent information in less prominent regions. Thus, it proposes an attention-driven contrastive learning framework that uses an attention-driven dynamic masking strategy that guides the model to focus on under-attended regions. The experiments show the proposed method improves previous methods.

**Strengths:**

- The paper is easy to follow.

- The motivation is interesting and the visualization is helpful.

**Weaknesses:**

- The improvement is slight, especially for part segmentation (only 0.1 for Ins. mIoU)

- As shown in Table 7, only when Mask Ratio is 0.6, the dynamic high-attention mask strategy achieves 92.3% accuracy, and other settings  get lower score and all of them get very close scores.

- As shown in Table 4, the low attention mask strategy achieve higher scores than Random Mask w.r.t. OBJ-ONLY. Please give reasonable explanation.

**Questions:**

Refer to Weaknesses

---

### Official Review · Reviewer_UF4r · 2024-11-03

**Soundness:** 3
**Presentation:** 3
**Contribution:** 3
**Rating:** 8
**Confidence:** 4

**Summary:**

This paper introduces a novel framework designed to improve the understanding of 3D point clouds through a combination of attention mechanisms and contrastive learning that can easily be plugged into other transformer-based architectures for point cloud processing.
According to the authors, transformer-based models for point cloud understanding are sensitive to perturbations due to a limited global comprehension as they mainly focus on more prominent regions and overlook under-attended (yet still important) regions. For this reason, they propose an Attention-Driven Dynamic Masking that directs the model to focus on under-attended regions by dynamically masking high-attention patches, and steering the network to have a more comprehensive understanding of global point cloud structures.
Furthermore, they propose to combine a Contrastive Learning Loss at pre-training time with the original loss of the main backbone to improve feature discrimination and generalization, as pre-training losses are mainly based on generative/reconstruction tasks.
Experiments show that PointACL’s masking strategy and loss configurations significantly enhance model performance.

**Strengths:**

1. In my opinion, the paper is well written and clear. The problem statement and the proposed solutions are well explained and presented. I particularly appreciated the plot showing the attended regions, showing that the proposed framework indeed learns a more global understanding of the input point cloud.
2. Results are good, and I think the main takeaway from this paper is that giving more importance to less prominent patches yields very robust results to perturbations.

**Weaknesses:**

At the moment, I do not see any critical weaknesses in the paper.

**Questions:**

1. Typo at line 245: regzions->regions.
2. I would like to ask the authors how the second part of Eq. 5 works. I understand that its goal is to give a perturbation to the probability distribution of the first part, but still, it is not clear to me how the second part affects the first part. Maybe the authors can clarify this aspect.
3. Are the hyper-parameters fixed across datasets/experiments or each experiment has its own hyper-parameters? For example, the authors in the ablation studies state that R=0.6 is the optimal mask ratio. Is this true for all datasets or the value has been changed for better performance? What about the Probability temperature and the Contrastive Loss weight?
4. Do the authors think that the proposed claims are valid only for transformer-based architecture? Would it be possible to apply similar strategies to simpler architectures such as a PointNet?

---

### Official Review · Reviewer_YSYw · 2024-11-03

**Soundness:** 2
**Presentation:** 3
**Contribution:** 2
**Rating:** 3
**Confidence:** 4

**Summary:**

This paper considers the sparse characteristics of point clouds that differ from other modalities such as text and images. Transformer-based models leverage the attention mechanism that focuses on significant areas while potentially disregarding less noticeable point patches, may lead to missing important underlying information. To address this, it introduces an innovative framework that integrates an attention-based dynamic masking strategy with contrastive learning. This approach aims to effectively capture both global and local features, while also improving robustness against perturbations and incomplete data.

**Strengths:**

1.	This paper is well-written and easy to read.

2.	The motivation that employing an attention-driven dynamic masking strategy to enhance the understanding of global structures within the point cloud is reasonable.

**Weaknesses:**

1.	Innovations are limited. The method simply generates an attention-guided masked point cloud, and adds the comparative loss of the original point patches and masked point patches on original loss function.

2.	Experimental results are not comprehensive.

•  Improvements over baseline methods are quite marginal and perhaps not statistically significant. While this method can be seamlessly integrated into mainstream transformer architectures, it remains unclear whether the performance gains are due to additional 300 epoch pre-training, an increase in parameters, or the dynamic masking strategy itself.

•  It has not achieved SOTA performance and is not compared with the latest methods，such as PointMamba [1], Point-FEMAE[2],  Point-GPT-L.

•  The author did not demonstrate the generality of the proposed method. Assuming that this dynamic masking strategy is effective, it should be applicable to all transformer-based point cloud models, such as Point BERT, Point-FEMAE, Point-GPT-L etc. The author lacks further analysis of the generality of the proposed ideas.

•  Is there any comparison of computational efficiency and cost with baseline?


[1] D. Liang, et al, “Pointmamba: A simple state space model for point cloud analysis,” in Advances in Neural Information Processing Systems, 2024.
[2] Y Zha, et al. Towards compact 3d representations via point feature enhancement masked autoencoders[C]//Proceedings of the AAAI Conference on Artificial Intelligence. 2024, 38(7): 6962-6970.

**Questions:**

The main questions have listed in the Weakness.

---

### Official Review · Reviewer_11qV · 2024-11-04

**Soundness:** 2
**Presentation:** 2
**Contribution:** 2
**Rating:** 3
**Confidence:** 5

**Summary:**

This paper introduces PointACL, a novel framework that addresses a critical limitation in Transformer-based point cloud understanding models - their tendency to overlook information in less prominent regions. The framework consists of two key components: (1) an attention-driven dynamic masking strategy that selectively masks high-attention regions, forcing the model to learn from under-attended areas, and (2) a contrastive learning approach that combines with original pre-training objectives to enhance feature discrimination and generalization. The method can be seamlessly integrated with existing architectures like Point-MAE and PointGPT.

**Strengths:**

1、The paper presents a novel and well-motivated solution by introducing attention-driven dynamic masking. Unlike previous works using fixed or random masking, PointACL adaptively masks high-attention regions to force learning from under-attended areas.
2、The paper provides extensive experimental evidence to support its claims, including thorough robustness analysis under various perturbations (Gaussian noise, rotation, scaling, point dropout).

**Weaknesses:**

（1）The paper lacks theoretical analysis or justification for why the attention-driven dynamic masking strategy works. There's no mathematical foundation explaining:
1、Why masking high-attention regions leads to better global understanding？
2、How the dynamic probability distribution affects learning？
3、What properties of the contrastive loss ensure better feature discrimination.
This theoretical gap makes it difficult to understand the method's fundamental principles and potentially limits its generalizability.
（2） The paper primarily compares with Point-MAE and PointGPT-S as backbones, but misses comparisons with other masking strategies from recent works.
（3）The performance improvement of this paper on few shot learning and part segmentation tasks is not significant.

**Questions:**

See Weaknesses.

---

### Note · Authors · 2024-11-13

I have read and agree with the venue's withdrawal policy on behalf of myself and my co-authors.